# A Survey on Evidential Deep Learning For Single-Pass Uncertainty Estimation

## Abstract

Popular approaches for quantifying predictive uncertainty in deep neural networks often involve a set of weights or models, for instance via ensembling or Monte Carlo Dropout. These techniques usually produce overhead by having to train multiple model instances or do not produce very diverse predictions. This survey aims to familiarize the reader with an alternative class of models based on the concept of *Evidential Deep Learning*: For unfamiliar data, they admit "what they don't know" and fall back onto a prior belief. Furthermore, they allow uncertainty estimation in a single model and forward pass by parameterizing *distributions over distributions*. This survey recapitulates existing works, focusing on the implementation in a classification setting. Finally, we survey the application of the same paradigm to regression problems. We also provide a reflection on the strengths and weaknesses of the mentioned approaches compared to existing ones and provide the most central theoretical results in order to inform future research.

## 1 Introduction

Many existing methods for uncertainty estimation leverage the concept of Bayesian Model Averaging, that approaches such as Monte Carlo (MC) Dropout (Gal & Ghahramani, 2016), Bayes-by-backprop (Blundell et al., 2015) or ensembling (Lakshminarayanan et al., 2017) can be grouped under (Wilson & Izmailov, 2020). This involves the approximation of an otherwise infeasible to compute integral using Monte Carlo samples – for instance from an auxiliary distribution or in the form of ensemble members. This implies the following problems: Firstly, the qual-

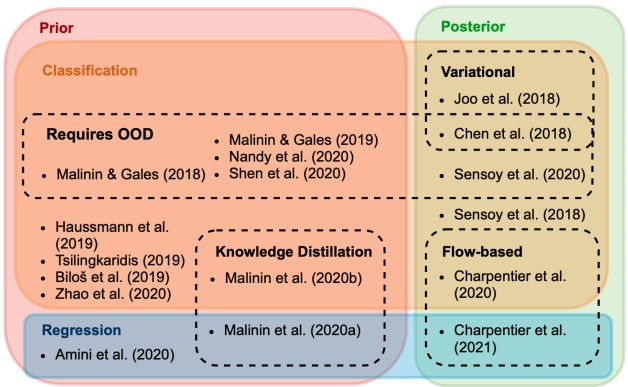

Figure 1: Taxonomy of surveyed approaches.

ity of the MC approximation depends on the veracity and diversity of samples from the weight posterior. Secondly, the approach often involves increasing the number of parameters in a model or training more model instances altogether. Recently, a new class of models has been proposed to side-step this conundrum by using a different factorization of the posterior predictive distribution. This allows to compute uncertainty in a single forward pass and set of weights. Furthermore, these models are grounded in a concept coined *Evidential Deep Learning*: For out-of-distribution (OOD) inputs, they fall back onto a prior, often expressed as *knowing what they don't know*.

Our contributions are as follows: We summarize the existing literature and group these approaches, critically reflecting on their advantages and shortcomings alike, as well as how they fare compared to other, related methods. This survey aims to both serve as an accessible introduction to this model family to the unfamiliar reader as well as an informative overview, in order to promote more applications outside the uncertainty estimation literature. We also provide a collection of the most important theoretical results for the Dirichlet distribution for Machine Learning, which plays a central role in many of the discussed approaches. We give an overview over all discussed work in Figure 1.

## 2 BACKGROUND

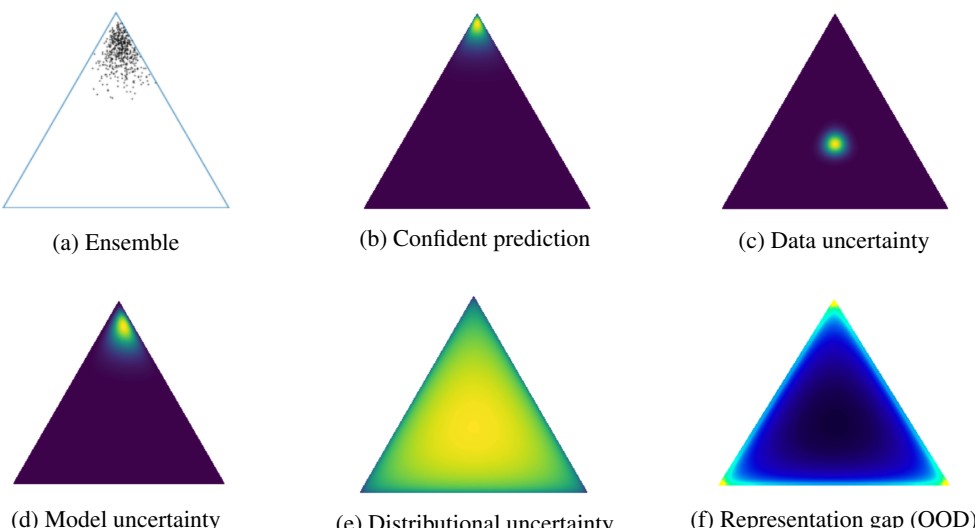

Figure 2: Examples of the probability simplex for a $K = 3$ classification problem, where every corner corresponds to a class and every point to a categorical distribution. Brighter colors correspond to higher density. (a) Ensemble of discriminators. (b) – (e) (Desired) Behavior of Dirichlet in different scenarios by Malinin & Gales (2018). (f) Representation gap by Nandy et al. (2020).

We first familiarize the reader with the necessary prerequisites for the rest of the survey, including Bayesian Model Averaging and the alternative approach of Evidential Deep Learning in Section 2.2 along with a short introduction to the Dirichlet distribution in the next section.

### 2.1 THE DIRICHLET DISTRIBUTION

The Beta distribution is a commonly used prior for a Bernoulli likelihood, which can be used to formulate a binary classification problem. The Dirichlet distribution arises as a multivariate generalization of the Beta distribution for a multi-class classification problem and is defined as follows:

$$\text{Dir}(\boldsymbol{\mu}; \boldsymbol{\alpha}) = \frac{1}{B(\boldsymbol{\alpha})} \prod_{k=1}^{K} \mu_k^{\alpha_k - 1}; \quad B(\boldsymbol{\alpha}) = \frac{\prod_{k=1}^{K} \Gamma(\alpha_k)}{\Gamma(\alpha_0)}; \quad \alpha_0 = \sum_{k=1}^{K} \alpha_k; \quad \alpha_k \in \mathbb{R}^+ \quad (1)$$

where $\Gamma(\cdot)$ denotes the gamma function, a generalization of the factorial to the real numbers, $K$ the number of categories or classes, and $B(\cdot)$ is called the beta function. For notational convenience, we also define $\mathbb{K} = \{1, \ldots, K\}$ as the set of all classes. The distribution is characterized by its *concentration parameters* $\boldsymbol{\alpha}$, the sum of which, often denoted as $\alpha_0$, is called the *precision*.[1] The distribution becomes relevant for applications using neural networks, considering that most modern networks for classification use a softmax function after their last layer to produce a categorical distribution of classes, for which the Dirichlet is a *conjugate prior*. The class probabilities can be expressed using a vector $\boldsymbol{\mu} \in [0, 1]^K$ s.t. $\mu_k \equiv P(y = k | x)$. Then, using a Dirichlet prior for a categorical likelihood, due to its conjugacy, produces a Dirichlet posterior with parameters $\boldsymbol{\beta}$, given a data set $\mathbb{D} = \{(x_i, y_i)\}_{i=1}^{N}$ of $N$ observations with corresponding labels:

---

[1]The precision is analogous to the precision of a Gaussian, where a larger $\alpha_0$ signifies a sharper distribution.

$$p(\boldsymbol{\mu}\,|\mathbb{D}, \boldsymbol{\alpha}) \propto p(\mathbb{D}|\,\boldsymbol{\mu})p(\boldsymbol{\mu}\,|\,\boldsymbol{\alpha}) = \prod_{i=1}^{N}\prod_{k=1}^{K}\mu_k^{\mathbf{1}_{y_i=k}} \frac{1}{B(\boldsymbol{\alpha})} \prod_{k=1}^{K}\mu_k^{\alpha_k-1}$$

$$= \prod_{k=1}^{K}\mu_k^{\left(\sum_{i=1}^{N}\mathbf{1}_{y_i=k}\right)} \frac{1}{B(\boldsymbol{\alpha})} \prod_{k=1}^{K}\mu_k^{\alpha_k-1} = \frac{1}{B(\boldsymbol{\alpha})} \prod_{k=1}^{K}\mu_k^{N_k+\alpha_k-1} = \mathrm{Dir}(\boldsymbol{\mu}; \boldsymbol{\beta}) \qquad (2)$$

where $\boldsymbol{\beta}$ is a vector with $\beta_k = \alpha_k + N_k$, with $N_k$ denoting the number of observations for class $k$ and $\mathbf{1}$ being the indicator function. Intuitively, this implies that the prior belief encoded by the initial Dirichlet is updated using the actual data, sharpening the distribution for classes for which many instances have been observed. This distribution constitutes a *distribution over categorical distributions* over the $K-1$ probability simplex, multiple instances of which are shown in Figure 2. Each point on the simplex corresponds to a categorical distribution, with the proximity to a corner indicating a high probability for the corresponding class. Figure 2a displays the predictions of an ensemble of classifiers as a point cloud on the simplex. Using a Dirichlet, this finite set of distributions can be extended to a continuous density over the whole simplex (Figures 2b to 2f).

## 2.2 Predictive Uncertainty in Neural Networks

In probabilistic modelling, uncertainty is commonly divided into aleatoric and epistemic uncertainty (Der Kiureghian & Ditlevsen, 2009; Hüllermeier & Waegeman, 2021). The former refers to the uncertainty that is induced by the data-generating process, and which e.g. might create an unresolvable overlap in class distributions.[2] The latter describes the uncertainty about the optimal model parameters (or even hypothesis class), reducible with an increasing amount of data, as less and less possible models become a plausible fit. These two notions resurface when formulating the posterior predictive distribution of a classifier for a new data point $\mathbf{x}$:[3]

$$p(y|\,\mathbf{x}) = \int \underbrace{P(y|\,\mathbf{x}, \boldsymbol{\theta})}_{\text{Aleatoric}} \underbrace{p(\boldsymbol{\theta}\,|\mathbb{D})}_{\text{Epistemic}} d\,\boldsymbol{\theta} \qquad (3)$$

For a large number of real-valued parameters $\boldsymbol{\theta}$ like in neural networks, this integral becomes intractable to evaluate, and thus is usually approximated using Monte Carlo samples – with the aforementioned problems of potential computational overhead and approximation errors. Malinin & Gales (2018) thus propose to factorize Equation (3) further:

$$p(y|\,\mathbf{x}) = \iint \underbrace{P(y|\,\boldsymbol{\mu})}_{\text{Aleatoric}} \underbrace{p(\boldsymbol{\mu}\,|\,\mathbf{x}, \mathbb{D})}_{\text{Distributional}} \underbrace{p(\boldsymbol{\theta}\,|\mathbb{D})}_{\text{Epistemic}} d\,\boldsymbol{\mu}\, d\,\boldsymbol{\theta} = \int P(y|\,\boldsymbol{\mu}) \underbrace{p(\boldsymbol{\mu}\,|\,\mathbf{x}, \hat{\boldsymbol{\theta}})}_{p(\boldsymbol{\theta}\,|\mathbb{D})=\delta(\boldsymbol{\theta}-\hat{\boldsymbol{\theta}})} d\,\boldsymbol{\mu} \qquad (4)$$

In the last step, we replace $p(\boldsymbol{\theta}\,|\mathbb{D})$ by a point estimate $\hat{\boldsymbol{\theta}}$ using the Dirac delta function, i.e. a single trained neural network, to get rid of the intractable integral. Although another integral remains, retrieving the uncertainty from this predictive distribution actually has a closed-form analytical solution for the Dirichlet (see Section 3.2). The advantage of this approach is further that it allows us to differentiate uncertainty about a data point because it lies in a region of considerable class overlap (Figure 2c) from it differing from the training distribution entirely (Figure 2e). Assuming that a point estimate of the parameters suffices prevents one from estimating epistemic uncertainty like in earlier works, as discussed in the next section. However, there are works like Haussmann et al. (2019); Zhao et al. (2020) that combine both approaches.

---

[2]Unless additional features are added.

[3]Note that the predictive distribution in Equation (3) recovers the common case for a single network prediction where $P(y|\,\mathbf{x}, \boldsymbol{\theta}) \approx P(y|\,\mathbf{x}, \hat{\boldsymbol{\theta}})$. Mathematically, this is expressed by replacing the posterior $p(\boldsymbol{\theta}\,|\mathbb{D})$ by a delta distribution like in Equation (4), where all probability density rests on a single parameter configuration.

## 3    DIRICHLET NETWORKS

We will show in Section 3.1 how neural networks can parameterize Dirichlet distributions, while Section 3.2 reveals how such parameterization can be exploited for efficient uncertainty estimation. The remaining sections enumerate different examples from the literature parameterizing either a prior (Section 3.3.1) or posterior Dirichlet distribution (Section 3.3.2) according to Equations (1) and (2).

### 3.1    PARAMETERIZATION

For a classification problem with $K$ classes, a neural classifier is usually realized as a function $f_{\boldsymbol{\theta}} : \mathbb{R}^D \to \mathbb{R}^K$, mapping to *logits* for each class given an input $\mathbf{x} \in \mathbb{R}^D$. Followed up by a softmax function, this then defines a categorical distribution over classes with a vector $\boldsymbol{\mu}$ s.t. $\mu_k \equiv p(y = k | \mathbf{x}, \boldsymbol{\theta})$. The same architecture can be used without any major modification to instead parameterize a *Dirichlet* distribution, as in Equation (1).[4] In order to classify a data point $\mathbf{x}$, a categorical distribution is created from the predicted concentration parameters of the Dirichlet as follows (this definition arises from the expected value, see Appendix A.1):

$$\boldsymbol{\alpha} = f_{\boldsymbol{\theta}}(\mathbf{x}); \quad \mu_k = \frac{\alpha_k}{\alpha_0}; \quad \hat{y} = \underset{k \in \mathbb{K}}{\arg \max} \ \mu_1, \dots, \mu_K \tag{5}$$

As discussed in Section 3.3.2, this process is very similar when parameterizing a Dirichlet posterior distribution, except that in this case, a term corresponding to the class observation in Equation (2) is added to every concentration parameter as well.

### 3.2    UNCERTAINTY ESTIMATION WITH DIRICHLET NETWORKS

Let us now turn our attention on how to estimate the different notions of uncertainty laid out in Section 2.2 within the Dirichlet framework. Although stated for the prior parameters $\boldsymbol{\alpha}$, the following methods can also be applied to the posterior parameters $\boldsymbol{\beta}$ as well without loss of generality.

**Data (aleatoric) uncertainty.**    For the data uncertainty, we can evaluate the expected entropy of the data distribution $p(y | \boldsymbol{\mu})$ (similar to previous works like e.g. Gal & Ghahramani, 2016). As the entropy captures the "peakiness" of the output distribution, a lower entropy indicates that the model is concentrating all probability mass on a single class, while high entropy stands for a more uniform distribution – the model thus is undecided about the right prediction. For Dirichlet networks, this quantity has a closed-form solution (for the full derivation, refer to Appendix B.1):

$$\mathbb{E}_{p(\boldsymbol{\mu} | \mathbf{x}, \hat{\boldsymbol{\theta}})} \left[ H \Big[ P(y | \boldsymbol{\mu}) \Big] \right] = - \sum_{k=1}^{K} \frac{\alpha_k}{\alpha_0} \left( \psi(\alpha_k + 1) - \psi(\alpha_0 + 1) \right) \tag{6}$$

where $\psi$ denotes the digamma function, defined as $\psi(x) = \frac{d}{dx} \log \Gamma(x)$, and $H$ the Shannon entropy.

**Model (epistemic) uncertainty.**    As we saw in Section 2.2, computing the model uncertainty in the classical sense via the weight posterior $p(\boldsymbol{\theta} | \mathbb{D})$ like in Blundell et al. (2015); Gal & Ghahramani (2016); Smith & Gal (2018) is usually not done in the Dirichlet framework (with exceptions such as Haussmann et al., 2019; Zhao et al., 2020). Nevertheless, the defining property of Dirichlet networks is that epistemic uncertainty is expressed in the spread of the Dirichlet distribution (for instance in Figure 2 (d) and (e)). Therefore, the epistemic uncertainty can be quantified considering the concentration parameters $\boldsymbol{\alpha}$ that shape this very same distribution: Charpentier et al. (2020) simply consider the maximum $\alpha_k$ as a score akin to the maximum probability score by Hendrycks & Gimpel, while Sensoy et al. (2018) compute it by $K / \sum_{k=1}^{K} (\alpha_k + 1)$ or simply $\alpha_0$ (Charpentier et al., 2020). In both cases, the underlying intuition is that larger $\alpha_k$ produce a sharper density, and thus indicate increased confidence in a prediction.

---

[4]The only thing to note here is that the every $\alpha_k$ has to be strictly positive, which can for instance be enforced by using an additional ReLU function (and adding a small value, e.g. like in Sensoy et al., 2020) on the output or predicting $\log \alpha_k$ instead (Sensoy et al., 2018; Malinin & Gales, 2018).

**Distributional uncertainty.** Another appealing property of this model family is to distinguish uncertainty due to model underspecification (Figure 2d) from uncertainty due to alien inputs (Figure 2e). In the Dirichlet framework, the distributional uncertainty can be deduced by computing the difference between the total amount of uncertainty and the data uncertainty, which can be expressed in terms of the mutual information between the label $y$ and its categorical distribution $\boldsymbol{\mu}$:

$$I\Big[y, \boldsymbol{\mu} \,\Big|\, \mathbf{x}, \mathbb{D}\Big] = \underbrace{H\Big[\mathbb{E}_{p(\boldsymbol{\mu} \,|\, \mathbf{x}, \mathbb{D})}\Big[P(y|\,\boldsymbol{\mu})\Big]\Big]}_{\text{Total Uncertainty}} - \underbrace{\mathbb{E}_{p(\boldsymbol{\mu} \,|\, \mathbf{x}, \mathbb{D})}\Big[H\Big[P(y|\,\boldsymbol{\mu})\Big]\Big]}_{\text{Data Uncertainty}} \tag{7}$$

Given that $\mathbb{E}[\mu_k] = \frac{\alpha_k}{\alpha_0}$ (Appendix A.1) and assuming the point estimate $p(\boldsymbol{\mu} \,|\, \mathbf{x}, \mathbb{D}) \approx p(\boldsymbol{\mu} \,|\, \mathbf{x}, \hat{\boldsymbol{\theta}})$ to be sufficient (Malinin & Gales, 2018), we obtain an expression very similar to Equation (6):

$$= -\sum_{k=1}^{K} \frac{\alpha_k}{\alpha_0}\bigg(\log \frac{\alpha_k}{\alpha_0} - \psi(\alpha_k + 1) + \psi(\alpha_0 + 1)\bigg)$$

### 3.3 EXISTING APPROACHES

The properties we discussed in previous sections are desirable traits, as they simplify the process of obtaining different uncertainty scores. However, it is important to note that the behaviors of the Dirichlet distributions in Figure 2 are idealized. In the empirical risk minimization framework that neural networks are usually trained in, Dirichlet networks are not incentivized to behave in the depicted way per se. Thus, when comparing existing approaches for parameterizing Dirichlet priors (Section 3.3.1) and posteriors (Section 3.3.2),[5] we mainly focus on the different ways that authors try to tackle this problem by means of loss functions and training procedures. Due to spatial constraints, we refrain to present all the different ideas in detail and instead only highlight some of them, summarizing the rest in an informal manner.[6] We give an overview over the discussed works in Tables 1 and 2 in the respective sections.

#### 3.3.1 PRIOR NETWORKS

The key challenge in training Dirichlet networks comes in the form of ensuring both high classification performance and the intended behavior under foreign data inputs. For this reason, most discussed works follow a loss function design using two parts: One optimizing for task accuracy for the former goal, the other one for a flat Dirichlet distribution for the latter.

As their main objective, Tsiligkaridis (2019) derive a generalized $l_p$ loss (see Appendix B.3), but using a local approximation of the Rényi divergence w.r.t to a uniform Dirichlet for the regularization term in order to ensure higher uncertainties for misclassified examples.[7] Zhao et al. (2020) similarly use a $l_2$ loss in the context of Graph Neural Networks (GNNs), but adapt a Kullback-Leibler (KL) regularization term to incorporate information about the local graph structure instead of referring to a uniform prior, as well as a knowledge distillation loss. Haussmann et al. (2019) optimize the model using a negative log-likelihood (NLL) loss and derive a regularizer from PAC bounds.

Instead of enforcing the flatness of the Dirichlet by itself, Malinin & Gales (2018) instead explicitly maximize the KL divergence to a uniform Dirichlet on OOD data points. Further, they instead utilize another KL term to train the model on predicting the correct label instead of a $l_p$ norm. However, as the KL divergence is not symmetrical, Malinin & Gales (2019) argue that the *reverse* counterparts of both loss terms actually have more appealing properties in producing the correct behavior of the predicted distribution (see Appendix B.5). Nandy et al. (2020) refine this idea further, stating that

---

[5]Even though the term *prior* and *posterior network* have been coined by Malinin & Gales (2018) and Charpentier et al. (2020) for their respective approaches, we use them in the following as an umbrella term for all methods targeting a prior or posterior Dirichlet.

[6]For more details, we refer the reader to Appendix A for general derivations concerning the Dirichlet distribution. We dedicate Appendix B to more extensive derivations of the different loss functions and regularizers and give a detailed overview over their mathematical forms in Appendix C.

[7]The Kullback-Leibler divergence can be seen as a special case of the Rényi divergence (van Erven & Harremoës, 2014), where the latter has a stronger information-threotic underpinning.

Table 1: Overview over prior networks for classification. (∗) OOD samples were created via temperature scaling inspired by Liang et al. (2018). ID: In-distribution.

| Method | Loss function | Architecture | Requires OOD samples? |
|---|---|---|---|
| Prior network (Malinin & Gales, 2018) | ID KL w.r.t smoothed label & OOD KL w.r.t. uniform prior | MLP / CNN | ✓ |
| Prior networks (Malinin & Gales, 2019) | Reverse KL of Malinin & Gales (2018) | CNN | ✓ |
| Information Robust Dirichlet Networks (Tsiligkaridis, 2019) | $l_p$ norm w.r.t one-hot label & Approx. Rényi divergence w.r.t. uniform prior | CNN | ✗ |
| Dirichlet via Function Decomposition (Biloš et al., 2019) | Uncertainty Cross-entropy & mean & variance regularizer | RNN | ✗ |
| Prior network with PAC Regularization (Haussmann et al., 2019) | Negative log-likelihood loss + PAC regularizer | BNN | ✗ |
| Ensemble Distribution Distillation (Malinin et al., 2020b) | Knowledge distillation objective | MLP / CNN | ✗ |
| Prior networks with representation gap (Nandy et al., 2020) | ID & OOD Cross-entropy + precision regularizer | MLP / CNN | ✓ |
| Prior RNN (Shen et al., 2020) | Cross-entropy + entropy regularizer | RNN | (✓)∗ |
| Graph-based Kernel Dirichlet distribution estimation (GKDE) (Zhao et al., 2020) | $l_2$ norm w.r.t. one-hot label & KL reg. with node-level distance prior & Knowledge distillation objective | GNN | ✗ |

even in this framework high epistemic and high distributional uncertainty (Figures 2d and 2e) might be confused, and instead propose novel loss functions producing a *representation gap* (Figure 2f; check Appendix C for the final form), which aims to be more easily distinguishable. Lastly, Malinin et al. (2020b) show that prior networks can also be distilled using an ensemble of classifiers and their predicted categorical distributions (akin to learning Figure 2e from Figure 2a), which does not require regularization at all (but training the ensemble).

An application to Natural Language Processing can be found in the work of Shen et al. (2020), who train their recurrent neural network for spoken language understanding using a simple cross-entropy loss and entropy regularizer. However, Biloš et al. (2019), who apply their model to asynchronous event classification, note that the standard cross-entropy loss only involves a point estimate of a categorical distribution, discarding all the information contained in the predicted Dirichlet. For this reason, they propose an *uncertainty-aware* cross-entropy (UCE) loss instead, which has a closed-form solution in the Dirichlet case (see Appendix B.6). They further regularize the mean and variance for OOD data points using an extra loss term.

### 3.3.2 POSTERIOR NETWORKS

Table 2: Overview over posterior networks for classification. OOD samples were created via (†) the fast-sign gradient method (Kurakin et al.) or (‡) using a Variational Auto-Encoder (VAE; Kingma & Welling, 2014). NLL: Negative log-likelihood. CE: Cross-entropy.

| Method | Loss function | Architecture | Requires OOD samples? |
|---|---|---|---|
| Evidential Deep Learning (Sensoy et al., 2018) | $l_2$ norm w.r.t. to one-hot label + KL w.r.t. uniform prior | CNN | ✗ |
| Variational Dirichlet (Chen et al., 2018) | ELBO + Contrastive Adversarial Loss | CNN | (✓)† |
| Belief Matching (Joo et al., 2020) | ELBO | CNN | ✗ |
| Posterior networks (Charpentier et al., 2020) | Uncertainty CE (Biloš et al., 2019) + Entropy regularizer | MLP / CNN + Norm. Flow | ✗ |
| Generative Evidential Neural Networks (Sensoy et al., 2020) | Contrastive NLL + KL between uniform & Dirichlet of wrong classes | CNN | (✓)‡ |

As elaborated on in Section 2.1, choosing a Dirichlet prior, due to its conjugacy to the categorical distribution, induces a Dirichlet posterior distribution. Like the prior in the previous section, this posterior can be parameterized by a neural network. The challenges hereby are two-fold: Accounting for the number of class observations $N_k$ that make up part of the posterior density parameters $\boldsymbol{\beta}$ (Equation (2)), and, similarly to prior networks, ensuring the wanted behavior on the probability simplex for in- and out-of-distribution inputs.

Sensoy et al. (2018) base their approach on the Dempster-Shafer theory of evidence (Yager & Liu, 2008; lending its name to the term "Evidential Deep Learning") and its formalization via subjective logic (Audun, 2018). In doing so, an agnostic belief in form of a uniform Dirichlet prior is updated using the aforementioned (pseudo-)counts, which are predicted by a neural network. Sensoy et al. (2018) train their model using a straightforward $l_2$ loss between the predicted Dirichlet and the one-hot encoded class label (Appendix B.4), as well as a regularization term consisting of the KL divergence w.r.t. a uniform Dirichlet:

$$\mathrm{KL}\big[p(\boldsymbol{\mu}\,|\,\boldsymbol{\alpha})\big|\big|p(\boldsymbol{\mu}\,|\,\mathbf{1})\big] = -\log\frac{\Gamma(K)}{B(\boldsymbol{\alpha})} + \sum_{k=1}^{K}(\alpha_k - 1)(\psi(\alpha_k) - \psi(\alpha_0))$$

In a follow-up work, Sensoy et al. (2020) train a similar model using a contrastive loss with artificial OOD samples from a Variational Autoencoder (Kingma & Welling, 2014), and a KL-based regularizer similar to that of Tsiligkaridis (2019). Charpentier et al. (2020) also set $\boldsymbol{\alpha}$ to a uniform prior, but obtain class observations $N_k$ from the training set and scale them by the probability of an input's latent representation $\mathbf{z}$ under a normalizing flow[8] (NF; Rezende & Mohamed, 2015) with parameters $\boldsymbol{\phi}$ and one flow instance per class (see Figure 3):

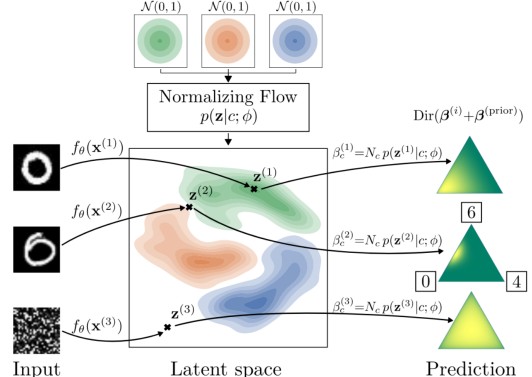

Figure 3: Schematic of a posterior network, taken from Charpentier et al. (2020). An encoder $f_{\boldsymbol{\theta}}$ maps inputs to a latent representation $\mathbf{z}$. NFs then model class-conditional densities, which are used together with the prior concentration to produce the posterior parameters.

$$\beta_k = \alpha_k + N_k \cdot p(\mathbf{z}\,|\,y=k,\boldsymbol{\phi}); \quad \mathbf{z} = f_{\boldsymbol{\theta}}(\mathbf{x})$$

This has the advantage of producing low probabilities for strange inputs like the noise in Figure 3, which in turn translate to low concentration parameters of the posterior Dirichlet, as it falls back onto the uniform prior. The model is then optimized using the same uncertainty-aware cross-entropy loss as in Biloš et al. (2019) with an additional entropy regularizer.

Another route lies in directly parameterizing the posterior parameters $\boldsymbol{\beta}$. Because it is infeasible to model the posterior this way due to an intractable integral, this leaves us to instead model an *approximate posterior* using variational inference methods, which is exactly the approach of Joo et al. (2020) and Chen et al. (2018). As the KL divergence between the true and approximate posterior is infeasible to estimate as well, the variational methods usually optimizes the *evidence lower bound* (ELBO) instead. For the Dirichlet family, the ELBO has an analytical solution (we refer the reader to Appendix A.3 for a derivation of the expression):

$$\mathcal{L}_{\mathrm{ELBO}} = \psi(\beta_y) - \psi(\beta_0) - \log\frac{B(\boldsymbol{\beta})}{B(\boldsymbol{\gamma})} + \sum_{k=1}^{K}(\beta_k - \gamma_k)\big(\psi(\beta_k) - \psi(\beta_0)\big)$$

---

[8] A NF is a deep generative model, estimating a density in the feature space by mapping it to a simple Gaussian in a latent space by a series of invertible, bijective transformations. The probability of an input can then be estimated by its latent encoding w.r.t. the simple Gaussian and the change-of-variable formula, traversing the flow in reverse. Instead of mapping from the feature space into latent space, the flows in Charpentier et al. (2020) map from the encoder latent space into a separate, second latent space.

## 4 EVIDENTIAL DEEP LEARNING FOR REGRESSION

Table 3: Overview over Evidential Deep Learning methods for regression.

| Method | Parameterized distribution | Loss function | Model |
|---|---|---|---|
| Deep Evidential Regression (Amini et al., 2020) | Normal-Inverse Gamma Prior | Negative log-likelihood loss + KL w.r.t. uniform prior | MLP / CNN |
| Regression Prior Network (Malinin et al., 2020a) | Normal-Wishart Prior | Reverse KL (Malinin & Gales, 2019) | MLP / CNN |
| Natural Posterior Network (Charpentier et al., 2021) | Inverse-$\chi^2$ Posterior | Uncertainty Cross-entropy (Biloš et al., 2019) + Entropy regularizer | MLP / CNN + Norm. Flow |

Because the Evidential Deep Learning framework provides such appealing properties, the question naturally arises of whether it can be extended to regression problems as well. The answer is yes, although the Dirichlet distribution is not an appropriate choice in this case. It is very common to model a regression problem using a normal likelihood (Bishop, 2006). As such, there are multiple potential choices for a prior distribution. The methods listed in Table 3 either choose the Normal-Inverse Gamma distribution (Amini et al., 2020; Charpentier et al., 2021), inducing a scaled inverse-$\chi^2$ posterior (Gelman et al., 1995),[9] as well as a Normal-Wishart prior (Malinin et al., 2020a). We will discuss these approaches in turn.

Amini et al. (2020) model the regression problem as a normal distribution with unknown mean and variance $\mathcal{N}(y; \mu, \sigma^2)$, and as such use a normal prior for the mean with $\mu \sim \mathcal{N}(\gamma, \sigma^2 v^{-1})$ and an inverse Gamma prior for the variance with $\sigma^2 \sim \Gamma^{-1}(\alpha, \beta)$, resulting in a combined Inverse-Gamma prior with parameters $\gamma, v, \alpha, \beta$. These are then predicted by different "heads" of a neural network. Aleatoric and epistemic uncertainty can then be estimated using the expected value of the variance as well as the variance of the mean, respectively, which have closed form solutions under this parameterization. The model is optimized using a negative log-likelihood objective along with an evidence regularizer, akin to the entropy one for Dirichlet networks. In the work of Charpentier et al. (2021), the authors generalize the approach behind the posterior networks by Charpentier et al. (2020) to different distributions from the exponential family, keeping architecture and loss function the same. Depending on the distributions used however, the UCE loss by Biloš et al. (2019) takes on a different form. Malinin et al. (2020a) can be seen as the multivariate generalization of the work of Amini et al. (2020), where a combined Normal-Wishart prior is formed to fit the now multivariate normal likelihood. Again, the prior parameters are the output of a neural network, and uncertainty can be quanitfied in a similar way. For training purposes, they apply the reverse KL objective of Malinin & Gales (2019) as well as the knowledge distillation objective of Malinin et al. (2020b).

## 5 RELATED WORK

The need for the quantification of uncertainty in order to earn the trust of end-users and stakeholders has been a key driver for research (Bhatt et al., 2021). Unfortunately, standard neural discriminator architectures have been proven to possess unwanted theoretical properties w.r.t. to OOD inputs[10] (Hein et al., 2019; Ulmer & Cinà, 2020) and lacking calibration in practice (Guo et al., 2017). A popular way to overcome these blemishes is by quantifying (epistemic) uncertainty by aggregating multiple predictions by networks in the Bayesian Model Averaging framework (Jeffreys, 1998; Wilson & Izmailov, 2020), Laplace approximations (Kristiadi et al., 2020; Daxberger et al., 2021), variational methods (Gal & Ghahramani, 2016; Blundell et al., 2015), ensembling (Lakshminarayanan et al., 2017) or mixtures of the latter two (Pearce et al., 2020; Wilson & Izmailov, 2020). Nevertheless, many of these methods have been shown not to produce diverse predictions (Wilson &

---

[9]The form of the Normal-Inverse Gamma posterior and the Normal Inverse-$\chi^2$ posterior are interchangable using some parameter substitutions (Murphy, 2007).

[10]Pearce et al. (2021) argue that some insights might partially be mislead by low-dimensional intuitions, and that empirically OOD data in higher dimensions tend to be mapped into regions of higher uncertainty.

Izmailov, 2020; Fort et al., 2019) and to deliver subpar performance and potentially misleading uncertainty estimates under distributional shift (Ovadia et al., 2019; Masegosa, 2020; Wenzel et al., 2020; Izmailov et al., 2021a;b), raising doubts about their efficacy.

The methods in Section 3.3 and Section 4 can be seen as single-pass alternatives that avoid approximating the predictive distribution in Equation (3) via Monte Carlo estimates. The proposed Posterior Network (Charpentier et al., 2020; 2021) can furthermore be seen as related to another, competing approach, namely the combination of neural discriminators with density estimation methods, for instance in the form of energy-based models (Grathwohl et al.; Elflein et al., 2021) or other hybrid architectures (Lee et al., 2018; Mukhoti et al., 2021).

Some of the discussed models have already found a variety of applications, such as in autonomous driving (Capellier et al., 2019; Liu et al., 2021; Petek et al., 2021), medical screening (Ghesu et al., 2019; Gu et al., 2021), molecular analysis (Soleimany et al., 2021), and open set recognition (Bao et al., 2021).

# 6 DISCUSSION

Despite their advantages, the last chapters have highlighted key weaknesses of Dirichlet networks as well: In order to achieve the right behavior of the distribution and thus guarantee sensible uncertainty estimates, some approaches Malinin & Gales (2018; 2019); Nandy et al. (2020); Malinin et al. (2020a) require out-of-distribution data points during training. This comes with two problems: Such data is often not available or in the first place, or cannot guarantee robustness against *other* kinds of unseen OOD data, of which infinite types exist in a real-valued feature space.[11] Indeed, Kopetzki et al. (2021) found OOD detection to deteriorate across a family of Dirichlet-based models under adversarial perturbation and OOD data points. One possible explanation for this behavior might lie in the insight that neural networks trained in the empirical risk minimization framework might learn spurious but highly predictive features (Ilyas et al., 2019; Nagarajan et al., 2021). This way, inputs stemming from the training distribution might be mapped to similar parts of the latent space as data points outside the distribution even though they have (from a human perspective) blatant semantic differences, simply because these semantic features were not useful to optimize for the training objective. This can result in ID and OOD points having assigned similar feature representations by a network, a phenomenon has been coined "feature collapse" (Nalisnick et al., 2019; van Amersfoort et al., 2021; Havtorn et al., 2021). One strategy to mitigate (but not solve) this issue has been to enforce a constraint on the smoothness of the neural network function (Wei et al., 2018; van Amersfoort et al., 2020; 2021; Liu et al., 2020), thereby maintaining both a sensitivity to semantic changes in the input and robustness against adversarial inputs (Yu et al., 2019). Nevertheless, this question remains an open area of research and the impact on Evidential Deep Learning methods underexplored.

# 7 CONCLUSION

This survey has given an overview over contemporary approaches for uncertainty estimation using neural networks to parameterize conjugate priors or the corresponding posteriors instead of likelihoods, with a focus on the Dirichlet distribution in a classification context. We highlighted their appealing theoretical properties allowing for uncertainty estimation with minimal computational overhead, rendering them as a viable alternative to existing strategies. We also emphasized practical problems: In order to nudge models towards the desired behavior in the face of unseen or out-of-distribution samples, the design of the model architecture and loss function have to be carefully considered. At the moment, the entropy regularizer seems to be a sensible choice in prior networks when OOD data is not available. Combining discriminators with generative models like normalizing flows like in (Charpentier et al., 2020; 2021), embedded in a sturdy Bayesian framework, also appears as an exciting direction for practical applications. In summary, we believe that recent advances show promising results for Evidential Deep Learning, making it a viable option in the realm of uncertainty estimation to improve safety and trustworthiness in Machine Learning systems.

---

[11]The same applies to the artificial OOD data in Chen et al. (2018); Shen et al. (2020); Sensoy et al. (2020).

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

# A  FUNDAMENTAL DERIVATIONS

This appendix section walks the reader through generalized versions of recurring theoretical results using Dirichlet distributions in a Machine Learning context, such as their expectation in Appendix A.1, their entropy in Appendix A.2 and the Kullback-Leibler divergence between two Dirichlets in Appendix B.3.

## A.1  EXPECTATION OF A DIRICHLET

Here, we show results for the quantities $\mathbb{E}[\mu_k]$ and $\mathbb{E}[\log \mu_k]$. For the first, one such derivation is given by Lin (2016), however we instead adapt the more elegant solution by Miller (2011):

$$\mathbb{E}[\mu_k] = \int \cdots \int \mu_k \frac{\Gamma(\alpha_0)}{\prod_{k=1}^{K} \Gamma(\alpha_k)} \prod_{k=1}^{K} \mu_k^{\alpha_k - 1} d\mu_1 \ldots d\mu_K$$

Moving $\mu_k^{\alpha_k - 1}$ out of the product:

$$= \int \cdots \int \frac{\Gamma(\alpha_0)}{\prod_{k=1}^{K} \Gamma(\alpha_k)} \mu_k^{\alpha_k - 1 + 1} \prod_{k' \neq k} \mu_{k'}^{\alpha_{k'} - 1} d\mu_1 \ldots d\mu_K \tag{8}$$

For the next step, we define a new set of Dirichlet parameters with $\beta_k = \alpha_k + 1$ and $\forall k' \neq k$ : $\beta_{k'} = \alpha_{k'}$. Now we can see that for those new parameters, $\beta_0$ is defined as $\beta_0 = \sum_k \beta_k = 1 + \alpha_0$. So by virtue of the Gamma function's property that $\Gamma(\alpha_0 + 1) = \alpha_0 \Gamma(\alpha_0)$, replacing all terms in the normalization factor will yield

$$= \int \cdots \int \frac{\alpha_k}{\alpha_0} \frac{\Gamma(\beta_0)}{\prod_{k=1}^{K} \Gamma(\beta_k)} \prod_{k=1}^{K} \mu_k^{\beta_k - 1} d\mu_1 \ldots d\mu_K = \frac{\alpha_k}{\alpha_0}$$

where in the last step we obtain the final result, since the Dirichlet with new parameters $\beta_k$ must nevertheless integrate to 1, and the integrals do not regard $\alpha_k$ or $\alpha_0$. For the expectation $\mathbb{E}[\log \mu_k]$, we first rephrase the Dirichlet distribution in terms of the exponential family (Kupperman, 1964). The exponential family encompasses many commonly-used distributions, such as the normal, exponential, Beta or Poisson, that all follow the form

$$p(\mathbf{x}; \boldsymbol{\eta}) = h(\mathbf{x}) \exp\left(\boldsymbol{\eta}^T u(\mathbf{x}) - A(\boldsymbol{\eta})\right)$$

with *natural parameters* $\boldsymbol{\eta}$, *sufficient statistic* $u(\mathbf{x})$, and *log-partition function* $A(\boldsymbol{\eta})$. For the Dirichlet distribution, Wikipedia (2021) provides the sufficient statistic as $u(\boldsymbol{\mu}) = [\log \boldsymbol{\mu}_1, \ldots, \boldsymbol{\mu}_K]^T$ and the log-partition function

$$A(\boldsymbol{\alpha}) = \sum_{k=1}^{K} \log \Gamma(\alpha_k) - \log \Gamma(\alpha_o) \tag{9}$$

By Wikipedia (2021), we also find that by the moment-generating function that for the sufficient statistic, its expectation can be derived by

$$\mathbb{E}[u(\mathbf{x})_k] = \frac{\partial A(\boldsymbol{\eta})}{\partial \eta_k} \tag{10}$$

Therefore we can evaluate the expected value of $\log \mu_k$ (i.e. the sufficient statistic) by inserting the definition of the log-partition function in Equation (9) into Equation (10):

$$\mathbb{E}[\log \mu_k] = \frac{\partial}{\partial \alpha_k} \sum_{k=1}^{K} \log \Gamma(\alpha_k) - \log \Gamma(\alpha_0) = \psi(\alpha_k) - \psi(\alpha_0) \tag{11}$$

which corresponds precisely to the definition of the digamma function as $\psi(x) = \frac{d}{dx} \log \Gamma(x)$.

## A.2 ENTROPY OF DIRICHLET

The following derivation is adapted from Lin (2016), with the result stated in Charpentier et al. (2020) as well.

$$
\begin{aligned}
H[\boldsymbol{\mu}] &= -\mathbb{E}[\log p(\boldsymbol{\mu} \mid \boldsymbol{\alpha})] \\
&= -\mathbb{E}\left[\log\left(\frac{1}{B(\boldsymbol{\alpha})} \prod_{k=1}^{K} \mu_k^{\alpha_k - 1}\right)\right] \\
&= -\mathbb{E}\left[-\log B(\boldsymbol{\alpha}) + \sum_{k=1}^{K}(\alpha_k - 1)\log\mu_k\right] \\
&= \log B(\boldsymbol{\alpha}) - \sum_{k=1}^{K}(\alpha_k - 1)\mathbb{E}[\log\mu_k]
\end{aligned}
$$

Using Equation (11):

$$
\begin{aligned}
&= \log B(\boldsymbol{\alpha}) - \sum_{k=1}^{K}(\alpha_k - 1)\big(\psi(\alpha_k) - \psi(\alpha_0)\big) \\
&= \log B(\boldsymbol{\alpha}) + \sum_{k=1}^{K}(\alpha_k - 1)\psi(\alpha_0) - \sum_{k=1}^{K}(\alpha_k - 1)\psi(\alpha_k) \\
&= \log B(\boldsymbol{\alpha}) + (\alpha_0 - K)\psi(\alpha_0) - \sum_{k=1}^{K}(\alpha_k - 1)\psi(\alpha_k)
\end{aligned}
$$

## A.3 KULLBACK-LEIBLER DIVERGENCE BETWEEN TWO DIRICHLETS

The following result is presented using an adapted derivation by Lin (2016) and appears in Chen et al. (2018) and Joo et al. (2020) as a starting point for their variational objective (see Appendix B.7). In the following we use $\mathrm{Dir}(\boldsymbol{\mu}; \boldsymbol{\alpha})$ to denote the optimized distribution, and $\mathrm{Dir}(\boldsymbol{\mu}; \boldsymbol{\gamma})$ the reference or target distribution.

$$
\begin{aligned}
\mathrm{KL}\Big[p(\boldsymbol{\mu}\mid\boldsymbol{\alpha})\big\|\,p(\boldsymbol{\mu}\mid\boldsymbol{\gamma})\Big] &= \mathbb{E}\left[\log\frac{p(\boldsymbol{\mu}\mid\boldsymbol{\alpha})}{p(\boldsymbol{\mu}\mid\boldsymbol{\gamma})}\right] = \mathbb{E}\left[\log p(\boldsymbol{\mu}\mid\boldsymbol{\alpha})\right] - \mathbb{E}\left[\log p(\boldsymbol{\mu}\mid\boldsymbol{\gamma})\right] \\
&= \mathbb{E}\left[-\log B(\boldsymbol{\alpha}) + \sum_{k=1}^{K}(\alpha_k - 1)\log\mu_k\right] \\
&\quad - \mathbb{E}\left[-\log B(\boldsymbol{\gamma}) + \sum_{k=1}^{K}(\gamma_k - 1)\log\mu_k\right]
\end{aligned}
$$

Distributing and pulling out $B(\boldsymbol{\alpha})$ and $B(\boldsymbol{\gamma})$ out of the expectation (they don't depend on $\boldsymbol{\mu}$):

$$
\begin{aligned}
&= -\log\frac{B(\boldsymbol{\gamma})}{B(\boldsymbol{\alpha})} + \mathbb{E}\left[\sum_{k=1}^{K}(\alpha_k - 1)\log\mu_k - (\gamma_k - 1)\log\mu_k\right] \\
&= -\log\frac{B(\boldsymbol{\gamma})}{B(\boldsymbol{\alpha})} + \mathbb{E}\left[\sum_{k=1}^{K}(\alpha_k - \gamma_k)\log\mu_k\right]
\end{aligned}
$$

Moving the expectation inward and using the identity $\mathbb{E}[\mu_k] = \psi(\alpha_k) - \psi(\alpha_0)$ from Appendix A.1:

$$
= -\log\frac{B(\boldsymbol{\gamma})}{B(\boldsymbol{\alpha})} + \sum_{k=1}^{K}(\alpha_k - \gamma_k)\big(\psi(\alpha_k) - \psi(\alpha_0)\big)
$$

The KL divergence is also used by some works as regularizier by penalizing the distance to a uniform Dirichlet with $\boldsymbol{\gamma} = \mathbf{1}$ (Sensoy et al., 2018). In this case, the result above can be derived to be

$$
\mathrm{KL}\Big[p(\boldsymbol{\mu}\,|\,\boldsymbol{\alpha})\big|\big|p(\boldsymbol{\mu}\,|\mathbf{1})\Big] = \log\frac{\Gamma(K)}{B(\boldsymbol{\alpha})} + \sum_{k=1}^{K}(\alpha_k - 1)\big(\psi(\alpha_k) - \psi(\alpha_0)\big)
$$

where the $\log\Gamma(K)$ term can also be omitted for optimization purposes, since it does not depend on $\boldsymbol{\alpha}$.

## B  ADDITIONAL DERIVATIONS

In this appendix we results relevant in a Machine Learning context. These include derivations of expected entropy (Appendix B.1) and mutual information (Appendix B.2) as uncertainty metrics for Dirichlet networks. Also, we derive a multitude of loss functions, including the $l_\infty$ norm loss of a Dirichlet w.r.t. a one-hot encoded class label in Appendix B.3, the $l_2$ norm loss in Appendix B.4, as well as the reverse KL loss by Malinin & Gales (2019), the UCE objective Biloš et al. (2019); Charpentier et al. (2020) and ELBO Shen et al. (2020); Chen et al. (2018) as training objectives (Appendices B.5 to B.7).

### B.1  DERIVATION OF EXPECTED ENTROPY

The following derivation is adapted from Malinin & Gales (2018) appendix section C.4.

$$
\mathbb{E}_{p(\boldsymbol{\mu}\,|\,\mathbf{x},\hat{\boldsymbol{\theta}})}\Big[H\big[P(y|\,\boldsymbol{\mu})\big]\Big] = \int p(\boldsymbol{\mu}\,|\,\mathbf{x},\hat{\boldsymbol{\theta}})\Big(-\sum_{k=1}^{K}\mu_k\log\mu_k\Big)d\boldsymbol{\mu}
$$
$$
= -\sum_{k=1}^{K}\int p(\boldsymbol{\mu}\,|\,\mathbf{x},\hat{\boldsymbol{\theta}})\Big(\mu_k\log\mu_k\Big)d\boldsymbol{\mu}
$$

Inserting the definition of $p(\boldsymbol{\mu}\,|\,\mathbf{x},\mathbb{D})$:

$$
= -\sum_{k=1}^{K}\left(\frac{\Gamma(\alpha_0)}{\prod_{k'=1}^{K}\Gamma(\alpha_{k'})}\int \mu_c\log\mu_c\prod_{k'=1}^{K}\mu_{k'}^{\alpha_{k'}-1}d\boldsymbol{\mu}\right)
$$

Singling out the factor $\mu_k$:

$$
= -\sum_{k=1}^{K}\left(\frac{\Gamma(\alpha_0)}{\Gamma(\alpha_k)\prod_{k'\in\mathbb{K}\setminus\{k\}}\Gamma(\alpha_{k'})}\int \mu_k\log\mu_k\prod_{k'\in\mathbb{K}\setminus\{k\}}\mu_{k'}^{\alpha_{k'}-1}\cdot\mu_k^{\alpha_k-1}d\boldsymbol{\mu}\right)
$$

Adjusting the normalizing constant (this is the same trick used in Appendix A.1):

$$
= -\sum_{k=1}^{K}\left(\frac{\alpha_c}{\alpha_0}\frac{\Gamma(\alpha_0 + 1)}{\Gamma(\alpha_k + 1)\prod_{k'\in\mathbb{K}\setminus\{k\}}\Gamma(\alpha_{k'})}\int \log\mu_k\prod_{k'\in\mathbb{K}\setminus\{k\}}\mu_{k'}^{\alpha_{k'}-1}\cdot\mu_k^{\alpha_k}d\boldsymbol{\mu}\right)
$$

Using the identity $\mathbb{E}[\log\mu_k] = \psi(\alpha_k) - \psi(\alpha_0)$ (Equation (11))

$$
= -\sum_{k=1}^{K}\frac{\alpha_k}{\alpha_0}\Big(\psi(\alpha_k + 1) - \psi(\alpha_0 + 1)\Big)
$$

We can accommodate the extra factor $\mu_k$ by adding 1 to its concentration parameter, adjusting the whole normalizing constant, and thus obtaining an expectation of $\log\mu_k$ w.r.t to a new Dirichlet distribution which includes this very factor. Therefore, the resulting Digamma functions are also adjusted to this new distribution.

## B.2 Derivation of Mutual Information

We start from the expression in Equation (7):

$$I\left[y, \boldsymbol{\mu} \,\Big|\, \mathbf{x}, \mathbb{D}\right] = H\left[\mathbb{E}_{p(\boldsymbol{\mu} \,|\, \mathbf{x}, \mathbb{D})}\left[P(y| \boldsymbol{\mu})\right]\right] - \mathbb{E}_{p(\boldsymbol{\mu} \,|\, \mathbf{x}, \mathbb{D})}\left[H\left[P(y| \boldsymbol{\mu})\right]\right]$$

Given that $\mathbb{E}[\mu_k] = \frac{\alpha_k}{\alpha_0}$ (Appendix A.1) and assuming that point estimate $p(\boldsymbol{\mu} \,|\, \mathbf{x}, \mathbb{D}) \approx p(\boldsymbol{\mu} \,|\, \mathbf{x}, \hat{\boldsymbol{\theta}})$ is sufficient (Malinin & Gales, 2018), we can identify the first term as the standard Shannon entropy $-\sum_{k=1}^K \mu_k \log \mu_k = -\sum_{k=1}^K \frac{\alpha_k}{\alpha_0} \log \frac{\alpha_k}{\alpha_0}$. Furthermore, the second part we already derived in Appendix B.1 and thus we obtain:

$$= -\sum_{k=1}^K \frac{\alpha_k}{\alpha_0} \log \frac{\alpha_k}{\alpha_0} + \sum_{k=1}^K \frac{\alpha_k}{\alpha_0}\left(\psi(\alpha_k + 1) - \psi(\alpha_0 + 1)\right)$$

$$= -\sum_{k=1}^K \frac{\alpha_k}{\alpha_0}\left(\log \frac{\alpha_k}{\alpha_0} - \psi(\alpha_k + 1) + \psi(\alpha_0 + 1)\right)$$

## B.3 $l_\infty$ Norm Derivation

In this section we follow elaborate on the derivation of Tsiligkaridis (2019) deriving a generalized $l_p$ loss, upper-bounding the $l_\infty$ loss. This in turn allows us to easily derive the $l_2$ loss used by Sensoy et al. (2018); Zhao et al. (2020). Here we assume the classification target $y$ to be provided in the form of a one-hot encoded label $\mathbf{y} = [\mathbf{1}_{y=1}, \ldots, \mathbf{1}_{y=K}]^T$.

$$\mathbb{E}_{p(\boldsymbol{\mu} \,|\, \mathbf{x}, \boldsymbol{\theta})}\left[|| \mathbf{y} - \boldsymbol{\mu} ||_\infty\right] \leq \mathbb{E}_{p(\boldsymbol{\mu} \,|\, \mathbf{x}, \boldsymbol{\theta})}\left[|| \mathbf{y} - \boldsymbol{\mu} ||_p\right]$$

$$\leq \left(\mathbb{E}_{p(\boldsymbol{\mu} \,|\, \mathbf{x}, \boldsymbol{\theta})}\left[|| \mathbf{y} - \boldsymbol{\mu} ||_p^p\right]\right)^{1/p}$$

$$= \left(\mathbb{E}[(1 - \mu_y)^p] + \sum_{k \neq y} \mathbb{E}[\mu_k^p]\right)^{1/p}$$

In order to compute the expression above, we first realize that all components of $\mu$ are distributed according to a Beta distribution Beta$(\alpha, \beta)$ (since the Dirichlet is a multivariate generalization of the beta distribution) for which the moment-generating function is defined as follows:

$$\mathbb{E}[\mu^p] = \frac{\Gamma(\alpha + p)\Gamma(\beta)\Gamma(\alpha + \beta)}{\Gamma(\alpha + p + \beta)\Gamma(\alpha)\Gamma(\beta)} = \frac{\Gamma(\alpha + p)\Gamma(\alpha + \beta)}{\Gamma(\alpha + p + \beta)\Gamma(\alpha)}$$

Given that the first term in Appendix B.3 is characterized by Beta$(\alpha_0 - \alpha_y, \alpha_y)$ and the second one by Beta$(\alpha_k, \alpha_0 - \alpha_k)$, we can evaluate it as follows:

$$\mathbb{E}_{p(\boldsymbol{\mu} \,|\, \mathbf{x}, \boldsymbol{\theta})}\left[|| \mathbf{y} - \boldsymbol{\mu} ||_\infty\right] \leq \left(\frac{\Gamma(\alpha_0 - \alpha_y + p)\Gamma(\alpha_0 - \cancel{\alpha_y} + \cancel{\alpha_y})}{\Gamma(\alpha_0 - \cancel{\alpha_y} + p + \cancel{\alpha_y})\Gamma(\alpha_0 - \alpha_y)} + \sum_{k \neq y} \frac{\Gamma(\alpha_k + p)\Gamma(\cancel{\alpha_k} + \alpha_0 - \cancel{\alpha_k})}{\Gamma(\cancel{\alpha_k} + p + \alpha_0 - \cancel{\alpha_k})\Gamma(\alpha_k)}\right)^{\frac{1}{p}}$$

$$= \left(\frac{\Gamma(\alpha_0 - \alpha_y + p)\Gamma(\alpha_0)}{\Gamma(\alpha_0 + p)\Gamma(\alpha_0 - \alpha_y)} + \sum_{k \neq y} \frac{\Gamma(\alpha_k + p)\Gamma(\alpha_0)}{\Gamma(p + \alpha_0)\Gamma(\alpha_k)}\right)^{\frac{1}{p}}$$

$$= \left(\frac{\Gamma(\alpha_0)}{\Gamma(\alpha_0 + p)}\right)^{\frac{1}{p}}\left(\frac{\Gamma\left(\sum_{k \neq y} \alpha_k + p\right)}{\Gamma\left(\sum_{k \neq y} \alpha_k\right)} + \sum_{k \neq y} \frac{\Gamma(\alpha_k + p)}{\Gamma(\alpha_k)}\right)^{\frac{1}{p}}$$

### B.4 $l_2$ NORM LOSS DERIVATION

Here we present an adapted derivation by Sensoy et al. (2018) for the $l_2$-norm loss to train Dirichlet networks. Here we again use an one-hot vector for a label with $\mathbf{y} = [\mathbf{1}_{y=1}, \ldots, \mathbf{1}_{y=K}]^T$.

$$\mathbb{E}_{p(\boldsymbol{\mu}\,|\,\mathbf{x},\boldsymbol{\theta})}\left[||\,\mathbf{y} - \boldsymbol{\mu}\,||_2^2\right] = \mathbb{E}\left[\sum_{k=1}^{K}(\mathbf{1}_{y=k} - \mu_k)^2\right] \tag{12}$$

$$= \mathbb{E}\left[\sum_{k=1}^{K}\mathbf{1}_{y=k}^2 - 2\mathbf{1}_{y=k}\mu_k + \mu_k^2\right] \tag{13}$$

$$= \sum_{k=1}^{K}\mathbf{1}_{y=k}^2 - 2\mathbf{1}_{y=k}\mathbb{E}[\mu_k] + \mathbb{E}[\mu_k^2] \tag{14}$$

Using the identity that $\mathbb{E}[\mu_k^2] = \mathbb{E}[\mu_k]^2 + \text{Var}(\mu_k)$:

$$= \sum_{k=1}^{K}\mathbf{1}_{y=k}^2 - 2\mathbf{1}_{y=k}\mathbb{E}[\mu_k] + \mathbb{E}[\mu_k]^2 + \text{Var}(\mu_k) \tag{15}$$

$$= \sum_{k=1}^{K}\left(\mathbf{1}_{y=k} - \mathbb{E}[\mu_k]\right)^2 + \text{Var}(\mu_k) \tag{16}$$

Finally, we use the result from Appendix A.1 and the result that $\text{Var}(\mu_k) = \frac{\alpha_k(\alpha_0 - \alpha_k)}{\alpha_0^2(\alpha_0+1)}$ (see Lin, 2016):

$$= \sum_{k=1}^{K}\left(\mathbf{1}_{y=k} - \frac{\alpha_k}{\alpha_0}\right)^2 + \frac{\alpha_k(\alpha_0 - \alpha_k)}{\alpha_0^2(\alpha_0+1)} \tag{17}$$

### B.5 DERIVATION OF REVERSE KL LOSS

Here we re-state and annotate the derivation of reverse KL loss by Malinin & Gales (2019) in more detail, starting form the forward KL loss by Malinin & Gales (2018).

$$\mathbb{E}_{p(\mathbf{x},y)}\left[\sum_{k=1}^{K}\mathbf{1}_{y=k}\text{KL}\Big[p(\boldsymbol{\mu}\,|\,\hat{\boldsymbol{\alpha}})\Big|\Big|p(\boldsymbol{\mu}\,|\,\mathbf{x},\boldsymbol{\theta})\Big]\right]$$

$$= \mathbb{E}_{p(\mathbf{x},y)}\left[\sum_{k=1}^{K}\mathbf{1}_{y=k}\int p(\boldsymbol{\mu}\,|\,\hat{\boldsymbol{\alpha}})\log\frac{p(\boldsymbol{\mu}\,|\,\hat{\boldsymbol{\alpha}})}{p(\boldsymbol{\mu}\,|\,\mathbf{x},\boldsymbol{\theta})}d\boldsymbol{\mu}\right]$$

Writing the expectation explicitly:

$$= \int\sum_{k=1}^{K}p(y=k,\mathbf{x})\sum_{k=1}^{K}\mathbf{1}_{y=k}\int p(\boldsymbol{\mu}\,|\,\hat{\boldsymbol{\alpha}})\log\frac{p(\boldsymbol{\mu}\,|\,\hat{\boldsymbol{\alpha}})}{p(\boldsymbol{\mu}\,|\,\mathbf{x},\boldsymbol{\theta})}d\boldsymbol{\mu}\,d\mathbf{x}$$

$$= \int\sum_{k=1}^{K}p(\mathbf{x})P(y=k|\,\mathbf{x})\sum_{k=1}^{K}\mathbf{1}_{y=k}\int p(\boldsymbol{\mu}\,|\,\hat{\boldsymbol{\alpha}})\log\frac{p(\boldsymbol{\mu}\,|\,\hat{\boldsymbol{\alpha}})}{p(\boldsymbol{\mu}\,|\,\mathbf{x},\boldsymbol{\theta})}d\boldsymbol{\mu}\,d\mathbf{x}$$

$$= \mathbb{E}_{p(\mathbf{x})}\left[\sum_{k=1}^{K}P(y=k|\,\mathbf{x})\sum_{k=1}^{K}\mathbf{1}_{y=k}\int p(\boldsymbol{\mu}\,|\,\hat{\boldsymbol{\alpha}})\log\frac{p(\boldsymbol{\mu}\,|\,\hat{\boldsymbol{\alpha}})}{p(\boldsymbol{\mu}\,|\,\mathbf{x},\boldsymbol{\theta})}d\boldsymbol{\mu}\right]$$

Adding factor in log, collapsing double sum:

$$= \mathbb{E}_{p(\mathbf{x})}\left[\sum_{k=1}^{K}P(y=k|\,\mathbf{x})\int p(\boldsymbol{\mu}\,|\,\hat{\boldsymbol{\alpha}})\log\left(\frac{p(\boldsymbol{\mu}\,|\,\hat{\boldsymbol{\alpha}})\sum_{k=1}^{K}P(y=k|\,\mathbf{x})}{p(\boldsymbol{\mu}\,|\,\mathbf{x},\boldsymbol{\theta})\sum_{k=1}^{K}P(y=k|\,\mathbf{x})}\right)d\boldsymbol{\mu}\right]$$

Reordering, separating constant factor from log:

$$
\begin{aligned}
= \mathbb{E}_{p(\mathbf{x})}\Bigg[ &\int \sum_{k=1}^{K} P(y=k|\mathbf{x}) p(\boldsymbol{\mu}|\hat{\boldsymbol{\alpha}}) \bigg( \log\bigg( \frac{\sum_{k=1}^{K} P(y=k|\mathbf{x}) p(\boldsymbol{\mu}|\hat{\boldsymbol{\alpha}})}{p(\boldsymbol{\mu}|\mathbf{x},\boldsymbol{\theta})} \bigg) \\
&- \underbrace{\log\Big( \sum_{k=1}^{K} P(y=k|\mathbf{x}) \Big)}_{=0} \bigg) d\boldsymbol{\mu} \Bigg] \\
= \mathbb{E}_{p(\mathbf{x})}\bigg[ &\mathrm{KL}\Big[ \underbrace{\sum_{k=1}^{K} P(y=k|\mathbf{x}) p(\boldsymbol{\mu}|\hat{\alpha})}_{\text{Mixture of } K \text{ Dirichlets}} \Big|\Big| p(\boldsymbol{\mu}|\mathbf{x},\boldsymbol{\theta}) \Big] \bigg]
\end{aligned}
$$

where we can see that this objective actually tries to minimizes the divergence towards a mixture of $K$ Dirichlet distributions. In the case of high data uncertainty, this is claimed incentivize the model to distribute mass around each of the corners of the simplex, instead of the desired behavior shown in Figure 2c. Therefore, Malinin & Gales (2019) propose to swap the order of arguments in the KL-divergence, resulting in the following:

$$
\mathbb{E}_{p(\mathbf{x})}\bigg[ \sum_{k=1}^{K} P(y=k|\mathbf{x}) \cdot \mathrm{KL}\Big[ p(\boldsymbol{\mu}|\mathbf{x},\boldsymbol{\theta}) \Big|\Big| p(\boldsymbol{\mu}|\hat{\boldsymbol{\alpha}}) \Big] \bigg]
$$

$$
= \mathbb{E}_{p(\mathbf{x})}\bigg[ \sum_{k=1}^{K} p(y=k|\mathbf{x}) \cdot \int p(\boldsymbol{\mu}|\mathbf{x},\boldsymbol{\theta}) \log \frac{p(\boldsymbol{\mu}|\mathbf{x},\boldsymbol{\theta})}{p(\boldsymbol{\mu}|\hat{\boldsymbol{\alpha}})} d\boldsymbol{\mu} \bigg]
$$

Reordering:

$$
= \mathbb{E}_{p(\mathbf{x})}\bigg[ \int p(\boldsymbol{\mu}|\mathbf{x},\boldsymbol{\theta}) \sum_{k=1}^{K} P(y=k|\mathbf{x}) \log \frac{p(\boldsymbol{\mu}|\mathbf{x},\boldsymbol{\theta})}{p(\boldsymbol{\mu}|\hat{\boldsymbol{\alpha}})} d\boldsymbol{\mu} \bigg]
$$

$$
= \mathbb{E}_{p(\mathbf{x})}\bigg[ \mathbb{E}_{p(\boldsymbol{\mu}|\mathbf{x},\boldsymbol{\theta})}\Big[ \sum_{k=1}^{K} P(y=k|\mathbf{x}) \log p(\boldsymbol{\mu}|\mathbf{x},\boldsymbol{\theta}) - \sum_{k=1}^{K} P(y=k|\mathbf{x}) \log p(\boldsymbol{\mu}|\hat{\boldsymbol{\alpha}}) \Big] \bigg]
$$

$$
= \mathbb{E}_{p(\mathbf{x})}\bigg[ \int p(\boldsymbol{\mu}|\mathbf{x},\boldsymbol{\theta}) \Big( \log\Big( \prod_{k=1}^{K} p(\boldsymbol{\mu}|\mathbf{x},\boldsymbol{\theta})^{P(y=k|\mathbf{x})} \Big) - \log\Big( \prod_{k=1}^{K} p(\boldsymbol{\mu}|\hat{\boldsymbol{\alpha}})^{P(y=k|\mathbf{x})} \Big) \Big) d\boldsymbol{\mu} \bigg]
$$

$$
= \mathbb{E}_{p(\mathbf{x})}\bigg[ \int p(\boldsymbol{\mu}|\mathbf{x},\boldsymbol{\theta}) \Big( \log\Big( p(\boldsymbol{\mu}|\mathbf{x},\boldsymbol{\theta})^{\sum_{k=1}^{K} P(y=k|\mathbf{x})} \Big)
$$

$$
- \log\Big( \prod_{k=1}^{K} \Big( \frac{1}{B(\boldsymbol{\alpha})} \prod_{k'=1}^{K} \mu_{k'}^{\alpha_{k'}-1} \Big)^{p(y=k|\mathbf{x})} \Big) \Big) d\boldsymbol{\mu} \bigg]
$$

$$
= \mathbb{E}_{p(\mathbf{x})}\bigg[ \int p(\boldsymbol{\mu}|\mathbf{x},\boldsymbol{\theta}) \Big( \log\big( p(\boldsymbol{\mu}|\mathbf{x},\boldsymbol{\theta}) \big) - \log\Big( \prod_{k=1}^{K} \Big( \frac{1}{B(\boldsymbol{\alpha})} \prod_{k'=1}^{K} \mu_{k'}^{\alpha_{k'}-1} \Big)^{P(y=k|\mathbf{x})} \Big) d\boldsymbol{\mu} \bigg]
$$

$$
= \mathbb{E}_{p(\mathbf{x})}\bigg[ \int p(\boldsymbol{\mu}|\mathbf{x},\boldsymbol{\theta}) \Big( \log\big( p(\boldsymbol{\mu}|\mathbf{x},\boldsymbol{\theta}) \big) - \log\Big( \frac{1}{B(\boldsymbol{\alpha})} \prod_{k'=1}^{K} \mu_{k'}^{\sum_{k=1}^{K} P(y=k|\mathbf{x})\alpha_{k'}-1} \Big) d\boldsymbol{\mu} \bigg]
$$

$$
= \mathbb{E}_{p(\mathbf{x})}\bigg[ \mathrm{KL}\Big[ p(\boldsymbol{\mu}|\mathbf{x},\boldsymbol{\theta}) || p(\boldsymbol{\mu}|\bar{\boldsymbol{\alpha}}) \Big] \bigg] \quad \text{where} \quad \bar{\boldsymbol{\alpha}} = \sum_{k=1}^{K} p(y=k|\mathbf{x})\alpha_{k'}
$$

Therefore, instead of a mixture of Dirichlet distribution, we obtain a single distribution whose *parameters are a mixture* of the concentrations of each class.

### B.6 UNCERTAINTY-AWARE CROSS-ENTROPY LOSS

The uncertainty-aware cross-entropy loss in Biloš et al. (2019); Charpentier et al. (2020) has the form

$$\mathcal{L}_{\text{UCE}} = \mathbb{E}_{p(\boldsymbol{\mu} \mid \mathbf{x}, \boldsymbol{\theta})}[\log p(y \mid \boldsymbol{\mu})] = \mathbb{E}[\log \mu_y] = \psi(\alpha_y) - \psi(\alpha_0)$$

as $p(y \mid \boldsymbol{\mu})$ is given by the true label in form of a delta distribution, we can apply the result from Appendix A.1.

### B.7 EVIDENCE-LOWER BOUND FOR DIRICHLET POSTERIOR ESTIMATION

The evidence lower bound is a well-known objective to optimize the KL-divergence between an approximate proposal and target distribution (Jordan et al., 1999; Kingma & Welling, 2014). We derive it based on Chen et al. (2018) in the following for the Dirichlet case with a proposal distribution $p(\boldsymbol{\mu} \mid \mathbf{x}, \boldsymbol{\theta})$ to the target distribution $p(\boldsymbol{\mu} \mid y)$. For the first part of the derivation, we omit the dependence on $\boldsymbol{\beta}$ for clarity.

$$\text{KL}\big[p(\boldsymbol{\mu} \mid \mathbf{x}, \boldsymbol{\theta}) \big| \big| p(\boldsymbol{\mu} \mid y)\big] = \mathbb{E}_{p(\boldsymbol{\mu} \mid \mathbf{x}, \boldsymbol{\theta})}\left[\log \frac{p(\boldsymbol{\mu} \mid \mathbf{x}, \boldsymbol{\theta})}{p(\boldsymbol{\mu} \mid y)}\right] = \mathbb{E}_{p(\boldsymbol{\mu} \mid \mathbf{x}, \boldsymbol{\theta})}\left[\log \frac{p(\boldsymbol{\mu} \mid \mathbf{x}, \boldsymbol{\theta})p(y)}{p(\boldsymbol{\mu}, y)}\right]$$

Factorizing $p(\boldsymbol{\mu}, y) = P(y \mid \boldsymbol{\mu})p(\boldsymbol{\mu})$, pulling out $p(y)$ as it doesn't depend on $\mu$:

$$= \mathbb{E}_{p(\boldsymbol{\mu} \mid \mathbf{x}, \boldsymbol{\theta})}\left[\log \frac{p(\boldsymbol{\mu} \mid \mathbf{x}, \boldsymbol{\theta})}{P(y \mid \boldsymbol{\mu})p(\boldsymbol{\mu})}\right] + p(y)$$

$$= \mathbb{E}_{p(\boldsymbol{\mu} \mid \mathbf{x}, \boldsymbol{\theta})}\left[\log \frac{p(\boldsymbol{\mu} \mid \mathbf{x}, \boldsymbol{\theta})}{p(\boldsymbol{\mu})}\right] - \mathbb{E}_{p(\boldsymbol{\mu} \mid \mathbf{x}, \boldsymbol{\theta})}\big[\log P(y \mid \boldsymbol{\mu})\big] + p(y)$$

$$\leq \text{KL}\big[p(\boldsymbol{\mu} \mid \mathbf{x}, \boldsymbol{\theta}) \big| \big| p(\boldsymbol{\mu})\big] - \mathbb{E}_{p(\boldsymbol{\mu} \mid \mathbf{x}, \boldsymbol{\theta})}\big[\log P(y \mid \boldsymbol{\mu})\big]$$

Now note that the second part of the result is the uncertainty-aware cross-entropy loss from Appendix B.6 and re-adding the dependence of $p(\mu)$ on $\boldsymbol{\gamma}$, we can re-use our result regarding the KL-divergence between two Dirichlets in Appendix A.3 and thus obtain:

$$\mathcal{L}_{\text{ELBO}} = \psi(\beta_y) - \psi(\beta_0) - \log \frac{B(\boldsymbol{\beta})}{B(\boldsymbol{\gamma})} + \sum_{k=1}^{K}(\beta_k - \gamma_k)\big(\psi(\beta_k) - \psi(\beta_0)\big) \tag{18}$$

which is exactly the solution obtained by both Chen et al. (2018) and Joo et al. (2020).

## C OVERVIEW OVER LOSS FUNCTIONS

In Tables 4 and 5, we compare the forms of the loss function used by Evidential Deep Learning methods for classification, using the consistent notation from the paper. Most of the presented results can be found in the previous Appendix A and Appendix B. We refer to the original work for details about the objective of Nandy et al. (2020).

Table 4: Overview over objectives used by prior networks for classification.

| Method | Loss function | Regularizer | Comment |
|---|---|---|---|
| Prior networks (Malinin & Gales, 2018) | $\log \frac{B(\hat{\boldsymbol{\alpha}})}{B(\boldsymbol{\alpha})} + \sum_{k=1}^{K}(\alpha_k - \hat{\alpha}_k)(\psi(\alpha_k) - \psi(\alpha_0))$ | $-\log \frac{\Gamma(K)}{B(\boldsymbol{\alpha})} + \sum_{k=1}^{K}(\alpha_k - 1)(\psi(\alpha_k) - \psi(\alpha_0))$ | Target concentration parameters $\hat{\boldsymbol{\alpha}}$ are created using a label smoothing approach, i.e. $\hat{\mu}_k = \begin{cases} 1 - (K-1)\varepsilon & \text{if } y = k \\ \varepsilon & \text{if } y \neq k \end{cases}$. Together with setting $\hat{\alpha}_0$ as a hyperparameter, $\hat{\alpha}_k = \hat{\mu}_k \hat{\alpha}_0$ |
| Prior networks (Malinin & Gales, 2019) | $\log \frac{B(\hat{\boldsymbol{\alpha}})}{B(\boldsymbol{\alpha})} + \sum_{k=1}^{K}(\alpha_k - \hat{\alpha}_k)(\psi(\alpha_k) - \psi(\alpha_0))$ | $\log \frac{B(\hat{\boldsymbol{\alpha}})}{B(\boldsymbol{\alpha})} + \sum_{k=1}^{K}(\alpha_k - \bar{\alpha}_k)(\psi(\alpha_k) - \psi(\alpha_0))$ | Similar to above, $\hat{\alpha}_c^{(k)} = \mathbf{1}_{c=k}\alpha_{\text{in}} + 1$ for in-distribution and $\bar{\alpha}_c^{(k)} = \mathbf{1}_{c=k}\alpha_{\text{out}} + 1$ where we have hyperparameters set to $\alpha_{\text{in}} = 0.01$ and $\alpha_{\text{out}} = 0$. Then finally, $\hat{\boldsymbol{\alpha}} = \sum_{k=1}^{K} p(y=k|\mathbf{x})\hat{\alpha}_k$ and $\bar{\boldsymbol{\alpha}} = \sum_{k=1}^{K} p(y=k|\mathbf{x})\bar{\alpha}_k$. |
| Information Robust Dirichlet Networks (Tsiligkaridis, 2019) | $\left(\frac{\Gamma(\alpha_0)}{\Gamma(\alpha_0 + p)}\right)^{\frac{1}{p}}\left(\frac{\Gamma\left(\sum_{k \neq y}\alpha_k + p\right)}{\Gamma\left(\sum_{k \neq y}\alpha_k\right)} + \sum_{k \neq y}\frac{\Gamma(\alpha_k + p)}{\Gamma(\alpha_k)}\right)^{\frac{1}{p}}$ | $\frac{1}{2}\sum_{k \neq y}(\alpha_k - 1)^2(\psi^{(1)}(\alpha_k) - \psi^{(1)}(\alpha_0))$ | $\psi^{(1)}$ is the polygamma function defined as $\psi^{(1)}(x) = \frac{d}{dx}\psi(x)$. |
| Dirichlet via Function Decomposition (Biloš et al., 2019) | $\psi(\alpha_y) - \psi(\alpha_0)$ | $\lambda_1 \int_0^T \mu_k(\tau)^2 d\tau + \lambda_2 \int_0^T (\nu - \sigma^2(\tau))^2 d\tau$ | Factors $\lambda_1$ and $\lambda_2$ that are treated as hyperparameters that weigh first term pushing the for logit $k$ to zero, while pushing the variance in the first term to $\nu$. |
| Prior network with PAC Reg. (Haussmann et al., 2019) | $-\log \mathbb{E}\left[\prod_{k=1}^{K}\left(\frac{\alpha_k}{\alpha_0}\right)^{\mathbf{1}_{k=y}}\right]$ | $\sqrt{\frac{\text{KL}\left[p(\boldsymbol{\mu}|\boldsymbol{\alpha})\middle|\middle|p(\boldsymbol{\mu}|\mathbf{1})\right] - \log \delta}{N}} - 1$ | The expectation in the loss function is evaluated using parameter samples from a weight distribution. $\delta \in [0, 1]$. |
| Ensemble Distribution Distillation (Malinin et al., 2020b) | $\psi(\alpha_0) - \sum_{k=1}^{K}\psi(\alpha_k) + \frac{1}{M}\sum_{m=1}^{M}\sum_{k=1}^{K}(\alpha_k - 1) \log p(y=k|\mathbf{x}, \boldsymbol{\theta}^{(m)})$ | - | The objective uses predictions from a trained ensemble with parameters $\boldsymbol{\theta}_1, \ldots, \boldsymbol{\theta}_M$. |
| Prior networks with representation gap (Nandy et al., 2020) | $-\log \mu_y - \frac{\lambda_{\text{in}}}{K}\sum_{k=1}^{K}\sigma(\alpha_k)$ | $-\sum_{k=1}^{K}\frac{1}{K}\log \mu_k - \frac{\lambda_{\text{out}}}{K}\sum_{k=1}^{K}\sigma(\alpha_k)$ | The main objective is being optimized on in-distribution, the regularizer on out-of-distribution data. $\lambda_{\text{in}}$ and $\lambda_{\text{out}}$ weighing terms and $\sigma$ denotes the sigmoid function. |
| Prior RNN (Shen et al., 2020) | $\sum_{k=1}^{K}\mathbf{1}_{k=y}\log \mu_k$ | $-\log B(\tilde{\boldsymbol{\alpha}}) + (\hat{\alpha}_0 - K)\psi(\hat{\alpha}_0) - \sum_{k=1}^{K}(\hat{\alpha}_k - 1)\psi(\hat{\alpha}_k)$ | Here, the entropy regularizer operates on a scaled version of the concentration parameters $\tilde{\boldsymbol{\alpha}} = (\mathbf{I}_K - \mathbf{W})\boldsymbol{\alpha}$, where $\mathbf{W}$ is learned. |
| Graph-based Kernel Dirichlet dist. est. (GKDE) (Zhao et al., 2020) | $\sum_{k=1}^{K}\left(\mathbf{1}_{y=k} - \frac{\alpha_k}{\alpha_0}\right)^2 + \frac{\alpha_k(\alpha_0 - \alpha_k)}{\alpha_0^2(\alpha_0 + 1)}$ | $-\log \frac{B(\boldsymbol{\alpha})}{B(\hat{\boldsymbol{\alpha}})} + \sum_{k=1}^{K}(\alpha_k - \hat{\alpha}_k)(\psi(\alpha_k) - \psi(\alpha_0))$ | $\hat{\boldsymbol{\alpha}}$ here corresponds to a uniform prior including some information about the local graph structure. The authors also use an additional knowledge distillation objective, which was omitted here since it doesn't related to the Dirichlet. |

Table 5: Overview over objectives used by posterior networks for classification.

| Method | Loss function | Regularizer | Comment |
|---|---|---|---|
| Evidential Deep Learning (Sensoy et al., 2018) | $\sum_{k=1}^{K} \left( \mathbf{1}_{y=k} - \frac{\alpha_k}{\alpha_0} \right)^2 + \frac{\alpha_k(\alpha_0-\alpha_k)}{\alpha_0^2(\alpha_0+1)}$ | $-\log \frac{\Gamma(K)}{B(\boldsymbol{\alpha})} + \sum_{k=1}^{K}(\alpha_k-1)(\psi(\alpha_k)-\psi(\alpha_0))$ | |
| Variational Dirichlet (Chen et al., 2018) | $\psi(\beta_y) - \psi(\beta_0)$ | $-\log \frac{B(\boldsymbol{\beta})}{B(\boldsymbol{\gamma})} + \sum_{k=1}^{K}(\beta_k-\gamma_k)(\psi(\beta_k)-\psi(\beta_0))$ | |
| Belief Matching (Joo et al., 2020) | $\psi(\beta_y) - \psi(\beta_0)$ | $-\log \frac{B(\boldsymbol{\beta})}{B(\boldsymbol{\gamma})} + \sum_{k=1}^{K}(\beta_k-\gamma_k)(\psi(\beta_k)-\psi(\beta_0))$ | |
| Posterior networks (Charpentier et al., 2020) | $\psi(\beta_y) - \psi(\beta_0)$ | $-\log B(\boldsymbol{\alpha}) + (\alpha_0-K)\psi(\alpha_0) - \sum_{k=1}^{K}(\alpha_k-1)\psi(\alpha_k)$ | |
| Generative Evidential Neural Network (Sensoy et al., 2020) | $-\sum_{k=1}^{K}\left( \mathbb{E}_{p_{in}(\mathbf{x})}\big[\log(\sigma(f_{\boldsymbol{\theta}}(\mathbf{x})))\big] + \mathbb{E}_{p_{out}(\mathbf{x})}\big[\log(1-\sigma(f_{\boldsymbol{\theta}}(\mathbf{x})))\big]\right)$ | $-\log \frac{\Gamma(K)}{B(\boldsymbol{\alpha}_{-y})} + \sum_{k\neq y}(\alpha_k-1)(\psi(\alpha_k)-\psi(\alpha_0))$ | The main loss is a discriminative loss using ID and OOD samples, generated by a VAE. The regularizer is taken over all classes *excluding* the true class $y$ (also indicated by $\boldsymbol{\alpha}_{-y}$). |