# OpenReview forum: "A Survey on Evidential Deep Learning For Single-Pass Uncertainty Estimation"
_ICLR.cc/2022/Conference — ICLR 2022 Submitted_

### Official Review · Reviewer_H7TT · 2021-10-29

**Correctness:** 3
**Technical Novelty And Significance:** 1
**Empirical Novelty And Significance:** Not applicable
**Recommendation:** 3
**Confidence:** 4

**Main Review:**

This is a survey paper without any conceptual claims of novelty. Its main purpose is to make an emerging family of machine learning models more accessible to the community. I find this goal sensible and important.

The paper presents the technical material very clearly and provides a well-justified dichotomy of the evidential deep learning model family.

I think the statement in Footnote 1 is a bit unnecessary confusion. The precision score of the Dirichlet distribution corresponds simply to the "precision" of a Gaussian distribution, which is customarily defined as the inverse of the covariance.

There exist few pieces of work that use Dirichlet priors on class probabilities, which are missed by this survey paper. For instance,

J. Gast and S. Roth, "Lightweight Probabilistic Deep Networks", CVPR, 2018

M. Haussmann et al., "Bayesian Evidential Deep Learning with PAC Regularization", AABI, 2020

The paper could clarify the qualitative difference between distributional uncertainty and representation gap better. It looks to me like they both quantify the same source of uncertainty: whether a test-time samples is coming from a different distribution than the training-time samples. They only differ in the quantitative aspect: how much the model is confident that a sample is out-of-domain. I am missing the rationale behind introducing a new category of uncertainty only for the sake of this quantitative difference.

The paper claims to collect some theoretical properties of the Dirichlet distribution that could be useful for machine learning. However, going through the paper and the appendix, I am not able to identify theoretical content that provides any new insights on the use of Dirichlet distribution for machine learning. The presented calculations seem to have been taken from the original papers in their vanilla form and some additional properties, such as expected L-infty norm and moment generating function are rather textbook material.

**Summary Of The Paper:**

The paper presents a survey of evidential deep learning, a family of machine learning methods that suggest accounting for various forms of uncertainties via a hierarchical predictive model whose hyperprior parameters are a function of input observations. The paper classifies evidential deep learning approaches into categories and points out the strengths and weaknesses of each category. It also presents some theoretical properties of the Dirichlet distribution that could be relevant for machine learning tasks.

**Summary Of The Review:**

Having said that the paper touches the important topic of making a new model family more accessible to the audience, I am afraid I do not think it does it in a way that would add sufficient value on top of one reading the material from the original papers. I think this way because the paper adopts most of the content verbatim from the original sources and does not add a level of abstraction that helps the reader draw new insights that are not straightforward from the original sources. Nor does it highlight overlooked positive or negative properties of the evidential model and nor does it present any empirical outcomes that one would find surprising. Under these conditions, I am not able to recommend an accept for this paper in its current shape.

---

> ### Author Response · Authors · 2021-11-18
> **Response to Reviewer H7TT**
>
> We thank reviewer H7TT for their constructive and favourable feedback. Many of the mentioned points were integrated into the updated draft. We are very appreciate, as they have improved the overall quality and completeness of our work. We now want to address the reviewer's main concerns: While we do adopt a lot of content of the discussed works, we try to do so by using a consistent notation and adding detailed annotations to derivations. We also see
> the contrasting of different approaches and embedding them into a common context as a valuable contribution to the community. Regarding the concerns of reviewer H7TT about not adding new insights to the summarized material, we refer to them to the response to reviewer vS5D, where we elaborate on this point in more detail.

---

> > ### Comment · Reviewer_H7TT · 2021-11-29
> > **Keep my score**
> >
> > Thanks for your response. While I do not mean to categorically reject a survey paper, I would expect from it a concrete value contribution to the community such as a new perspective or surprising empirical outcomes. As they remain untouched, I keep my original score.

---

### Official Review · Reviewer_vS5D · 2021-10-29

**Correctness:** 4
**Technical Novelty And Significance:** 1
**Empirical Novelty And Significance:** Not applicable
**Recommendation:** 1
**Confidence:** 3

**Main Review:**

In my opinion, the paper did a very good job of collecting papers on evidential deep learning and  providing a brief overview of their ideas. As a person unfamiliar with this topic, I got interested and proceeded to read some of the references as I didn't fully grasp the motivation and all the details from a rather short introduction. The paper could have been more enjoyable if it wasn't limited to 9 pages and was written in a form of a tutorial with more intuition for all the equations. For example, it took me some time to interpret Eq.4,6,7. What bothers me about evidential deep learning is that the choice of the regularization term often seems arbitrary. I wish this paper had some deeper insights on  this matter.

I haven't seen survey papers published at conferences before, so I'm hesitant to recommend its acceptance. It would have been different if the paper, in addition to being a survey, presented some fundamental insights into the nature of these methods, which could count as a novel contribution.

I think it can become a well-cited journal or arxiv paper that can get people interested in the topic of evidential deep learning if the background sections are turned into a self-contained tutorial, and the survey parts include some additional motivation and intuition regarding the design choices that each of the papers make.

Extra remarks:
- should Eq.2 be $p(\mu | D, \alpha)$ instead of $p(\alpha | D, \mu)$?
- section 3.2 typo: quanitified
- section 3.3.2 typo: uncertaint-aware





**Summary Of The Paper:**

The paper is a survey of methods in evidential deep learning. It gives a brief motivation for this set of methods, explains a general framework, and describes previous works for classification and regressions tasks in varying amount of details. As a survey paper, it doesn't introduce novel ideas, but gives a useful overview for people who would like to start working on this topic.

**Summary Of The Review:**

Contributions of this paper are not novel, which is normal for a survey paper.
I think survey papers are not suitable for conferences as they need to be evaluated using a different set of criteria.
Therefore, my recommendation would be to reject this paper purely based on its type.

---

> ### Author Response · Authors · 2021-11-18
> **Reponse to Reviewer vS5D**
>
> We thank reviewer vS5D for their favourable comments regarding our work. We have fixed the mentioned issues in
> the updated draft.
>
> We now respond to the doubts of reviewers 8cR8, vS5D, H7TT regarding the format of this paper
> and its appropriateness here jointly, since it was the main point of reviewer vS5D:
>
> We are aware to publishing reviews or survey like ours at top-tier venues is unusual, but could set a precedent.
> Despite not providing any new empirical or theoretical results, we believe that the value of our work in lies in the following:
>
> 1.) Providing a overview over the existing approaches, familiarizing readers with this new model class. We added a paragraph about practical applications to underline the manifold of potential applications.
> 2.) Providing a comprehensive resource over important theoretical results in order to support novel users in their adoption. While we do not state any new results, we state them in a consistent form and in one single resource, instead of being scattered across many works.
> 3.) Critically reflecting on the shortcomings and disadvantages of this model class in comparison with other, similar approaches. Such a characterization is often omitted from other works in order to solely highlight strengths and sway reviewers towards positive scores.
>
> We argue that especially in a quickly-moving field like Machine Learning, normalizing the publishing of papers such as this work can help to consolidate knowledge, a view that as especially inspired by the opinion paper by [1].
> In their study, [2] furthermore show how an accelerated publishing environment tends to ossify a field's canon, making it harder for new ideas to gain traction, an effect that we would like to mitigate with this work by disseminating ideas about these methods in a bundled way. In short, we believe that this style of paper is complementary to novel and SOTA-improving approaches in its value to the Machine Learning community.
>
> [1] Lampinen, Andrew Kyle, et al. "Publishing fast and slow: A path toward generalizability in psychology and AI." (2021).
> [2] Chu, Johan SG, and James A. Evans. "Slowed canonical progress in large fields of science." Proceedings of the National Academy of Sciences 118.41 (2021).

---

### Official Review · Reviewer_8cR8 · 2021-11-02

**Correctness:** 4
**Technical Novelty And Significance:** 1
**Empirical Novelty And Significance:** 1
**Recommendation:** 5
**Confidence:** 4

**Main Review:**

Why is Sensoy et al. 2018 called ‘prior networks’ (it seems the original papers did not referred to their method as prior network either). According to the equation in Section 3.3.1, shouldn’t the prior be the flat Dirichlet? What is needed to be trained is more like an approximate posterior network that takes as input a new data x. It would be helpful for readers to elaborate on this.

In Equation 5, should $\mu_0, …, \mu_K$ be $\mu_1, …, \mu_K$? Otherwise, $\mu_0$ will always be the largest by definition, which makes on sense.

For Equation 6, is some term related to $B(\alpha)$ missing?

The argument that the model uncertainty can simply replaced by some function of $\alpha$ is not very convincing, given that the network will directly output $\alpha$. It would be helpful if more details and intuition can be provided.

As a survey, this manuscript seems to miss a lot of important related work on probabilistic neural networks and Bayesian deep learning [a-f].

Minor:

P6: fig. 2 -> Fig. 2



[a] Natural-Parameter Networks: A Class of Probabilistic Neural Networks, NIPS 2016
[b] Feed-forward Propagation in Probabilistic Neural Networks with Categorical and Max Layers, ICLR 2018
[c] Sampling-free Epistemic Uncertainty Estimation Using Approximated Variance Propagation, CVPR 2019
[d] Probabilistic Backpropagation for Scalable Learning of Bayesian Neural Networks, ICML 2015
[e] Being Bayesian, Even Just a Bit, Fixes Overconfidence in ReLU Networks, ICML 2020
[f] A Survey on Bayesian Deep Learning, ACM Computing Surveys, 2020


**Summary Of The Paper:**

This paper surveys a collection of existing works that the author frames as evidential deep learning. For the classification case, evidential deep learning tries to train a network to output the parameters of a Dirichlet distribution, hoping the one could directly obtain the data uncertainty and model uncertainty from some functions of the output parameters. The authors provide a relatively comprehensive review of recent work in this direction. Besides classification, evidential deep learning for regression is also discussed briefly.

**Summary Of The Review:**

Overall, the paper is informative and did a relatively good job in introducing the basic concepts as well as the motivation of evidential deep learning. However, it seems there are still some important related works that are missing from the references. Besides, the paper in its current form looks more like a review rather than a comprehensive thorough survey, partly because of the space constraints in ICLR. Therefore I am not sure that ICLR is the right venue for it. Perhaps it is more suitable as a long journal where more details and taxonomy could be included.

---

> ### Author Response · Authors · 2021-11-18
> **Reponse to Reviewer 8cR8**
>
> We thank reviewer 8cR8 for their feedback and notes on the paper. We included many of their notes in the updated draft and extended our references to Bayesian Deep Learning. Below we address some specific points:
>
> - "For Equation 6, is some term related to missing?": As the derivation in Appendix section B.1 shows, the final expression is obtained by converting the expression into a new Dirichlet distribution with a new normalizing constant, allowing us to use a previously derived identity for the expectation. As such, the normalizing term $B(\alpha)$ drops out. This step is also explained in more detail in appendix section A.1.
>
> - "The argument that the model uncertainty can simply replaced by some function of alpha is not very convincing": In order to make this point more intuitive, we would like to draw a comparison to the logits in a vanilla classifier.
> Popular uncertainty metrics in this case as maximum probability [1] or predictive entropy [2] operate on the softmax-normalized logits, and thus uncertainty is expressed as a function of the network output as well. The reviewer might rightfully respond that uncertainty is not only expressed via network outputs, but also via the (distribution of) model weights. While in the usual framework we would quantify this uncertainty using multiple models, via ensembling, MC Dropout or other techniques, the distribution over distributions is encoded by $\alpha$ directly
> (see fig. 2a and b-e). While $\alpha$ and logits are somewhat analogous, the former has a direct probabilistic interpretation. We would be more than happy to engage in a discussion with the reviewer should we have mischaracterized or misunderstood their point.
>
> Lastly, regarding doubts of the appropriateness of such a review for ICLR, we refer the reviewer to the response to reviewer vS5D, where this points will be elaborated on in more detail.
>
> [1] Hendrycks, Dan, and Kevin Gimpel. "A baseline for detecting misclassified and out-of-distribution examples in neural networks." arXiv preprint arXiv:1610.02136 (2016).
> [2] Gal, Yarin. "Uncertainty in deep learning." (2016): 3.

---

> ### Comment · Reviewer_8cR8 · 2021-11-29
> **Thanks for the clarification**
>
> I have read the authors’ response as well as other reviewers’ comments. While I appreciate the response, I agree with other reviewers on the lack of concrete technical contribution as a conference paper.
>
> My general take is that there is certainly value in this informative survey and the authors might want to consider submitting the survey to more suitable venues such as related journals. However, as the other reviewers pointed out given the relatively smaller number of papers in the direction, this is more like a (short) review paper than a (full) ‘survey’ paper.
>
> Based on the reasons above, I would like to keep my score unchanged.

---

### Official Review · Reviewer_eCYd · 2021-11-03

**Correctness:** 3
**Technical Novelty And Significance:** 2
**Empirical Novelty And Significance:** Not applicable
**Recommendation:** 5
**Confidence:** 4

**Main Review:**

**Strengths**:
* In general, the paper is well written, structured, and easy to understand.
* There is significant value in surveying evidential deep learning techniques as there has been some parallel work in this space that is not often connected. This survey covers and categorizes a large number of papers under the umbrella of evidential deep learning.
* Fig. 1 is an excellent visualization and categorization of approaches in this space.
* I liked the classification of the literature into prior and posterior methods throughout the survey.

**Weaknesses**:
* A significant portion of the survey repeats ideas from the Prior Network paper (Malinin and Gales, 2018) with some typos (e.g., in Eqs. 3 and 4, the condition on the dataset as per the Prior Network paper has been omitted). I am missing further insight and takeaways beyond the existing collection of papers surveyed. The discussion details some potential avenues for future work, and there are some insights in the footnotes. It would be good to bring these footnote insights into the paper and expound on them (particularly since there is enough space). However, as it stands, the survey does not provide sufficient additional insight or context for the ideas presented to form its contribution.
* The representation gap (Nandy et al., 2020) and its resulting visualization were not described in sufficient detail.
* Overall, it appears that there was insufficient material for a 9 page paper as equations were extremely spaced out (e.g., Eqs. 3 and 4), and even then the paper did not reach the page limit, making it look unpolished.
* In general, the presentation of the paper needs improvement. There were frequent typos (e.g., 'We also to provide', 'e. 4', 'uncertaint-aware') and inconsistent formatting for referencing figures, sections, and equations (i.e., capitalization and abbreviation differences). There were some notational issues, as well. For example, it was confusing to see the definition $\mu_k = P(y = k \mid x, \theta)$, with and without $\theta$, and then to have the distribution $p(y \mid \mu)$. Furthermore, the references list is unpolished: repeating conference names, listing urls, etc.

**Suggestions for improvement**:
* My main recommendation is to work into the survey further insights and context from beyond the surveyed papers. For example, the origin of the term *evidential deep learning* comes from evidential theory [1], which relates to the Dirichlet distribution through the ideas in subjective logic [2]. It would be good to further highlight that prior networks collapse the uncertainty over the model parameters, as this insight is important and worth spending more space on in the paper.
* The Sensoy et al. AAAI paper [3] is missing from the survey, and should be incorporated.
* Additional investigation into the advantages and disadvantages of the different metrics used to evaluate the performance of evidential deep learning methods with respect to OOD inputs and uncertainty calibration would be valuable.
* Since this field does not have an unreasonable number of papers, the authors should consider running some of their own experiments to provide empirical conclusions, comparing methods that were not already compared in existing work (e.g., (Charpentier et al., 2020) and (Sensoy et al., 2020)).
* Lastly, the paper should be polished and proofread. I recommend the use of the cleveref package in LaTeX with the 'capitalise' option to avoid inconsistencies in referencing figures, equations, and sections.

[1] A. P. Dempster. A generalization of Bayesian inference. Classic works of the Dempster-Shafer Theory of Belief Functions, pages 73–104, 2008.

[2] A. Jøsang. Subjective Logic: A formalism for reasoning under uncertainty. Springer, 2016.

[3] M. Sensoy, et al. Uncertainty-Aware Deep Classifiers Using Generative Models. In AAAI, 2020.

**Summary Of The Paper:**

The proposed paper surveys recent work in evidential deep learning, and provides some potential directions for future work in this space. Evidential deep learning is a set of techniques for calibrating uncertainty of neural networks through second-order distributions.

**Summary Of The Review:**

The idea of a survey on evidential deep learning is excellent and valuable to researchers. However, in its current state, I do not believe the paper is ready for publication at ICLR. With improvements to the broader insights and takeaways from the survey as well as paper presentation, this work will become a good contribution to the academic community.

---

> ### Author Response · Authors · 2021-11-18
> **Response to Reviewer eCYd**
>
> We thank Reviewer eCYd for their elaborate and constructive feedback. We highly appreciate their various notes, as they undoubtedly improved the content and presentation of the paper. Most of the mentioned improvements were incorporated into the updated draft. We acknowledge that empirical comparisons between the different methods as suggested by the reviewer would certainly yield interesting insights, but are out of scope, as they would likely constitute their own paper. The format of our work is intended to be more of a meta review-style nature, a point that we elaborate on in the response to reviewer vS5D.

---

### Public Comment · ~Meet_P._Vadera1 · 2021-11-16
**Useful survey**

Hi,

This is a useful survey! We have some past work around compressing Bayesian posterior distribution for uncertainty estimation using distillation with experiments involving downstream applications like OOD detection. We think this might be relevant to this paper [1]- hope you find it useful!

1. Meet P. Vadera, Brian Jalaian, and Benjamin M. Marlin. Generalized bayesian posterior expectation distillation for deep neural networks. In Proceedings of the 36th Conference on Uncertainty in Artificial Intelligence (UAI), volume 124 of Proceedings of Machine Learning Research, pages 719–728. PMLR, 03–06 Aug 2020b. URL https://proceedings.mlr.press/v124/ vadera20a.htm
To view the comment, click here: https://openreview.net/forum?id=NE8B5RQkau&noteId=QO0F-McKts

---

### Decision · Program_Chairs · 2022-01-20

**Decision:**

Reject

**Comment:**

This paper surveys a collection of existing works that the author frames as evidential deep learning.

While the paper has been recognized as a nicely written survey, all reviewers have raised the major concern that the paper does not have a sufficient academic contribution compared to the surveyed papers. In particular, novelty appears to be limited as the paper does not offer novel views into the surveyed subfield.

Given the strong consensus among reviewers, I recommend rejecting this paper.